# Multi-Mycotoxin Long-Term Monitoring Survey on North-Italian Maize over an 11-Year Period (2011–2021): The Co-Occurrence of Regulated, Masked and Emerging Mycotoxins and Fungal Metabolites

**DOI:** 10.3390/toxins14080520

**Published:** 2022-07-29

**Authors:** Sabrina Locatelli, Valentina Scarpino, Chiara Lanzanova, Elio Romano, Amedeo Reyneri

**Affiliations:** 1Council for Agricultural Research and Economics (CREA), Research Centre for Cereal and Industrial Crops, Via Stezzano, 24, 24126 Bergamo, Italy; chiara.lanzanova@crea.gov.it; 2Department of Agricultural, Forest and Food Sciences (DISAFA), Università degli Studi di Torino, Largo Braccini 2, 10095 Grugliasco, Italy; valentina.scarpino@unito.it (V.S.); amedeo.reyneri@unito.it (A.R.); 3Council for Agricultural Research and Economics (CREA), Research Centre for Engineering and Agro-Food Processing, Via Milano 43, 24047 Treviglio, Italy; elio.romano@crea.gov.it

**Keywords:** aflatoxin B_1_, *Aspergillus*, deoxynivalenol, fumonisins, *Fusarium*, survey, zearalenone

## Abstract

Maize is considered one of the most susceptible crops to mycotoxin-producing fungi throughout the world, mainly belonging to the *Fusarium* spp. and *Aspergillus* spp. Maize is mainly used as animal feeds in Italy, as well as for human consumption, being essential for all the protected designation of origin (DOP) products. Our study investigated the occurrence of regulated mycotoxins in 3769 maize grain samples collected from 88 storage centers by the National Monitoring Network over an 11-year period (2011–2021). Moreover, an in-depth survey over a 4-year period, characterized by extremely different meteorological conditions, was conducted to investigate the co-occurrence of regulated, masked, and emerging mycotoxins. The survey confirmed that *Fusarium* spp. was the most frequent fungi and fumonisins were the main mycotoxins that were constantly detected in the different years and areas. Moreover, the areas characterized by high fumonisin levels were also the most prone to contamination by emerging mycotoxins produced by the same *Fusarium* species of the *Liseola* section. On the other hand, as a result of climatic changes, maize grains have also been affected by the increased frequency of aflatoxin accumulation. Deoxynivalenol, zearalenone, and other emerging mycotoxins produced by the same *Fusarium* species as the *Discolor* section occurred more abundantly in some areas in Northern Italy and in years characterized by predisposing meteorological conditions.

## 1. Introduction

Maize (*Zea mays*) is the most important grain and forage crop in Italy. It is mainly used as an energy ingredient in livestock feeds, but its utilization in many food products and bioindustrial derivatives is increasing to a great extent [1]. Maize grains are subject to infection from several fungal diseases of the *Aspergillus*, *Fusarium*, and *Penicillium* genera. The colonization and growth of these fungi can lead to an altered grain quality, and this mainly occurs through mycotoxin contamination [2]. 

Because of the frequent occurrence and negative impact of mycotoxins on human and animal health, several countries have established mycotoxin regulations for food and feeds. In the European Union (EU), maximum concentrations of aflatoxin B_1_ (AF B_1_), deoxynivalenol (DON), zearalenone (ZEA), ochratoxin A (OTA), and the sum of fumonisin B_1_ (FB_1_) and B_2_ (FB_2_) are in place for foodstuffs, while guidance levels have been introduced for T2-HT2 [3,4]. As of yet, maximum limits for feedstuffs are only in force for AF B_1_, while guidance levels have been set for DON, ZEN, OTA and the sum of FB_1_ and FB_2_ [5,6]. Currently, the EU regulations do not take into account the occurrence of masked and emerging mycotoxins or secondary fungal metabolites and the presence of multiple mycotoxins. Although mycotoxin regulations globally focus on a few mycotoxins individually, maize grains are frequently contaminated simultaneously by several different mycotoxins, including conjugated and emerging mycotoxins. 

Maize, due to its sensitivity to infection by toxigenic molds and to the ecological conditions during the growth and ripening of the grain, is the crop that is the most sensitive to high mycotoxin contamination in Southern Europe [7]. This is particularly evident in the Po Valley, where 95% of all the Italian grain production is concentrated. The maize grown in the plains of Northern Italy is cultivated in an environment with high air humidity and high temperatures, which are extremely favorable for the development of the main toxigenic molds. In fact, high contaminations of mycotoxins frequently occur in commercial lots under these environmental conditions [8]. For this reason, over the last two decades, thanks to the increasing attention paid to safety aspects by cereal supply chains, it has been found that mycotoxin contamination is actually the first cause of the decrease in the market value of maize grains [9]. Furthermore, a frequent and severe contamination by mycotoxins has been indicated as one of the main causes of a significant contraction of the maize cultivation area in Italy; in the past 2 decades, 50% of the acreage has been lost and converted for the cultivation of crops less prone to contamination, such as soybean or small cereals.

In the past decades, several surveys have been published around the world, including in Northern Italy, to highlight mycotoxin contamination profiles [7,10,11,12,13,14]. However, only a few surveys have investigated the mycotoxin profiles of regulated mycotoxins, masked, emerging mycotoxins, and secondary fungal metabolites [15,16]. Moreover, no other study has focused on maize production in Northern Italy by analyzing 3769 grain samples from 88 storage centers over an 11-year period.

Thus, the aims of this survey were twofold. The first aim was to draw up mycotoxin profiles of the maize grain grown in Northern Italy to highlight the different occurrences between years and production areas in order to point out the environmental conditions that aggravate the contaminations of regulated mycotoxins. The second aim was to investigate the role of emerging mycotoxins and their co-occurrence with the regulated ones.

## 2. Results

### 2.1. National Maize Monitoring Network

Our study investigated the occurrence of regulated mycotoxins in maize grain samples collected by the Italian National Monitoring Network, supported by the Italian Ministry of Agriculture, over an 11-year period (2011–2021). This network, which is coordinated by the CREA Research Centre for Cereal and Industrial Crops, is composed of 88 maize storage centers distributed throughout five macro-geographical areas (Figure 1). The Po Valley is the main maize growing area in Italy. 

### 2.2. Metereological Trends

We analyzed the different meteorological trends observed over the 2011–2021 period by measuring the daily mean precipitation and temperature during the ripening period of maize kernels (Figure 2A). 

The rainfall range observed over the years, where the average of 10 meteorological stations was considered, was between 152.08 mm in 2011 and 350.33 mm in 2014. Differences were also observed regarding temperatures. The minimum, 20.75 °C, was observed during 2014, while the maximum, 23.76 °C, was noted during 2018. The 2015 growing season also showed high temperatures (23.46 °C), which were counterbalanced by low rainfall (155.62 mm) with respect to 2018 (209.83 mm). As can be seen in Figure 2, the 2014 growing period underwent both more abundant rainfall (350.33 mm) and lower temperatures (20.75 °C) than the other years. The 2017 and 2021 growing seasons also showed low temperature levels of 22.49 °C and 22.53 °C, respectively.

The mean meteorological trend was also observed considering the five geographical areas (Figure 2B). The precipitation and temperature data from the two meteorological stations chosen to represent each geographical area were averaged over the 2011–2021 years.

A different distribution of rainfall was observed within the different areas. The W, SP, and A areas appear to be the areas with the lowest levels of precipitation, with 142.75 mm (W), 126.09 mm (SP), and 170.85 mm (A), respectively. However, these geographical areas also showed substantial differences in temperature. The W area was in fact cooler than the SP and A ones. The mean temperature for W was 21.86 °C, while it was 22.69 °C and 23.09 °C for the SP and A areas, respectively. C and E were instead the geographical areas in which the highest mean precipitation levels were observed: 260.31 mm for C and 305.51 mm for E.

### 2.3. Regulated Mycotoxins

The occurrence of FBs and AF B_1_ in maize lots is summarized in Table 1. FBs were the most common mycotoxins, being present in at least 97% of the samples analyzed from 2011 to 2021. The statistical analysis showed that the years with the highest and lowest contaminations were 2019 and 2011. During these years, the contamination level ranged between 25 and 47,300 µg/kg. By analyzing the spread in different geographical areas, E appeared to be most contaminated while W was the cleanest over the years.

Throughout the years in which this study was carried out, the presence of AF B_1_ was lower than that of FBs, with an average level of between 0.6 in 2014 and 12.5 µg/kg in 2012.

In 2014, the cleanest year, AF B_1_ was only detected in 7% of the samples, while in 2019, the most infected year, 59% of them showed an AF B_1_ level above 1 µg/kg. The statistical analysis showed that the areas of greatest incidence were SP and A while the W area was the least contaminated.

The effects of the year and area on the occurrence of DON and ZEA in the maize lots were also detected (Table 2).

The years with the higher levels of contamination were 2013 and 2014 while the year 2021 was found to be the least contaminated. 

In 2014, the maximum concentration reached 36,583 µg/kg, and 100% of the analyzed samples were positive.

When analyzing the geographical areas, W and A showed the highest and lowest DON concentration values.

Similarly, 2014 was the most contaminated year for ZEA, with an average value of 364 µg/kg, a maximum value of 1395 µg/kg, and 91% of positive samples; 2017 was the cleanest year. When considering the spread in different areas, W and E were found to be the most contaminated. The other areas had low levels of contamination, ranging from 43 µg/kg in C to 21 µg/kg in A.

The year with the lowest incidence and contamination of each mycotoxin, which we referred to as the best year, was compared with the worst year, namely the one with the highest incidence and contamination. The diffusion of regulated mycotoxins in different maize lots in Northern Italy, as obtained from the drying and storage centres, was also determined (Figure 3).

The year 2019 resulted to be the worst one for the incidence of FBs. All the areas had percentages of samples with FB concentrations above 4000 µg/kg, which is the maximum level for maize for human food use [3], and they ranged from a minimum of 67% in the SP area to 89% in the W ones. In 2011, the best year, the percentages of samples above the maximum level set in the EU were lower, that is, over a range of 1% in W to 18% in E (Figure 3A). 

In the best years, such as 2014, the incidence of AF B_1_ was insignificant: from 97% in the W area to 100% in the C, SP and A areas, and the samples had a lower AF B_1_ concentration than 2 µg/kg. In 2012, in which AF B_1_ was widespread on maize, 50% of the samples from area A had a larger AF B_1_ concentration than 20 µg/kg, thereby far exceeding the maximum level of 8 µg/kg set by the European Commission (EC) for AF B_1_ for maize intended for human consumption [4] (Figure 3B). 

DON and ZEA showed low incidences in the maize samples analyzed over the years. In 2021, 100% of the samples, in all the areas, had a DON content of less than 750 µg/kg; in 2017, the detected ZEA concentration in all the samples was below 350 µg/kg, that is, the maximum level set by the EC for maize for food purposes [3]. The year 2014 was the worst year for both mycotoxins in all the growing areas. DON had a greater concentration than 1750 µg/kg, the maximum level set by the EC for maize for human consumption [3], and ranged between 22% and 79% in the samples from areas A and W, respectively (Figure 3C). In the same areas, ZEA showed a greater concentration than 350 µg/kg, with values of 6% and 59%, respectively (Figure 3D).

Figure 4 shows the mean contamination level of FBs in maize lots from drying and storage centers located in different areas during the 2011–2021 period. According to the statistical analysis, the E area on average showed the highest level of FB accumulation in all the areas. Instead, the area with the lowest level of FBs was generally the W one. The years with the greatest accumulation of FBs were 2012, 2013, and 2019, while 2011 was the least-contaminated year.

The mean contamination levels of AF B_1_, regarding the same maize lots reported for FBs, during the 2012–2021 period, are shown in Figure 5. According to the statistical analysis, the A and SP areas on average showed higher AF B_1_ accumulation levels than all the other areas. The area with lowest level of AF B_1_ was generally the W area. The year 2012 showed the greatest accumulation of AF B_1_ in the different maize growing areas, except in the W area, due to the meteorological conditions that occurred, which favored AF B_1_ production by *Aspergillus* spp. In fact, except for a few years and in some geographical areas, the level of AF B_1_ was on average lower that that recorded in 2012.

The mean contamination level of DON is shown in Figure 6. According to the statistical analysis, 2013 and 2014 were the worst years, with the greatest accumulation of DON of all the years. In both years, the W and A areas were the most and the least contaminated, respectively. The best growing season, with the lowest levels of DON contamination, was 2021.

Figure 7 shows the mean contamination level of ZEA in the investigated maize lots during the 2014–2021 period. According to the statistical analysis, 2014 was the most contaminated year and W area showed higher levels of ZEA accumulation than all the other geographical areas. The A area was on average the cleanest one. The best year for ZEA accumulation was 2017.

Figure 8 shows the spread in different geographic areas for each mycotoxin, in the year of the highest incidence, considering the different classes of incidence established by Regulations (EC) n. 1126/2007 and n. 165/2010 [3,4].

In the worst year for FBs (2019, Figure 8A), almost the entire geographical area showed levels above 4000 µg/kg, except for a small area in E, where the levels were lower (1000–4000 µg/kg). 

During 2012, the worst year for aflatoxin accumulation (Figure 8B), the lowest levels of AF B_1_ were found in W and some areas in North E and C (0–5 µg/kg). The other areas showed distributions that fell into the highest classes. In the worst year for DON incidence (2014, Figure 8C), the SP (near Bologna) and A areas showed concentration levels included in the lowest classes, 0–1750 µg/kg. The W, C, and E areas and part of SP showed DON levels above the limit of 1750 µg/kg. A similar distribution was observed for ZEA (Figure 8D) with a lower level of incidence than that of DON.

Figure 9 shows the mean concentration of each mycotoxin detected during the investigated years. FBs (Figure 9A) were found to be endemically present in different geographical areas, with average distributed levels of between 1000 and >4000 µg/kg. As far as the AF B_1_ distribution is concerned, the average levels were generally distributed over the 0–10 µg/kg classes. Some small areas, where the average accumulation was greater (10–20 µg/kg), are highlighted in the figure. The distributions of DON and ZEA over the years generally showed an average distribution over all the considered incidence classes.

A Principal Component Analysis (PCA) was carried out to investigate the relationships between the main detected regulated mycotoxins, as it allows the maize samples to be clustered as similar mycotoxin contaminations for the different years and areas. The results of the PCA performed with the regulated mycotoxins detected over eleven growing seasons and in five maize areas in Northern Italy were used to reduce the number of variables to only 2, the 2 first principal components (PCs) producing a loading plot in which the two PCs, respectively, explained 39% (Principal Component 1, PC1) and 21% (Principal Component 2, PC2) of the total variations of the maize lot samples (Figure 10). The first component differentiated the years (Figure 10A) and the areas (Figure 10B) according to their mycotoxin contents, as produced by *F. graminearum* and *F. culmorum*, DON and ZEA, (loadings >|0.84|), and was also positively related to the rainfall during the ripening period and negatively related to the mean temperature during the ripening period. On the other hand, the second component differentiated the years (Figure 10A) and the areas (Figure 10B) according to their mycotoxin contents, as produced by *F. verticillioides* and *F. proliferatum*, FBs, and by *Aspergillus* spp. AF B_1_ (loadings > |0.71|). Overall, the PCA confirmed that the samples characterized by high contaminations of DON and ZEA were mainly clustered in the W area in 2014, while the SP and A areas as well as the 2012 (data not shown because not all the mycotoxins were analyzed for 2012), 2015, 2016, and 2017 years were more prone to AF B_1_ contamination. Furthermore, although FBs were constantly present in the different areas over the years, the E area and the 2019 year were the most contaminated ones.

### 2.4. Masked Mycotoxins

An in-depth study was carried out during the 2012–2015 period to evaluate the contamination levels of masked and modified mycotoxins, and emerging fungal metabolites in the same maize lots. As far as the masked mycotoxins are concerned, the mainly detected forms were those associated with DON, such as deoxynivalenol-3-glucoside (DON-3-G), and the acetylated forms, 3-acetyldeoxynivalenol (3-ADON) and 15-acetyldeoxynivalenol (15-ADON) (Table 3). 

Overall, the highest contaminations were detected for DON-3-G, which on average reached about the 27% of the DON contamination level detected in the same period during the 2012–2015 period, and this was followed by 15-ADON, which showed 12% of the DON content, while 3-ADON represented only 2% of the DON contamination. The 2014 year was the most contaminated for both DON-3-G and 15-ADON and was significantly different from the other years. On the other hand, the 2013 year, which was the second-most-contaminated year, after 2013, for DON-3-G and 15-ADON, was also the most contaminated for 3-ADON. 

As far as the areas are concerned, area W was significantly more prone to the contamination of all the masked forms of DON than the other areas, as previously reported for the “native” DON form (Table 3).

### 2.5. Emerging Mycotoxins and Other Fungal Metabolites

The emerging mycotoxins and fungal metabolites mainly produced by *Fusarium* spp. of the *Liseola* section, such as beauvericin (BEA), bikaverin (BIK), fusaric acid (FA), fusaproliferin (FUS), and moniliformin (MON), were detected for all the years and in all the areas. The effects of year and area on the occurrence of these emerging mycotoxins and fungal metabolites in maize lots from drying and storage centers located in different areas in Northern Italy during the 2012–2015 period are reported in Table 4. 

The 2013 year was the one with the highest level of contamination for all the detected fungal metabolites produced by *Fusarium* spp. of the *Liseola* section, with the exception of MON, for which 2012 was significantly more contaminated than the other years. As far as the areas are concerned, no significant differences among the different areas were recorded for any of the fungal metabolites reported in Table 4, with the exception of BIK and FA, for which area A recorded a significantly higher contamination than the other areas. 

Moreover, the emerging mycotoxins and fungal metabolites mainly produced by *Fusarium* spp. of the *Discolor* section, such as aurofusarin (AUR), butenolide (BUT), culmorin (CULM), and nivalenol (NIV), were also detected for all the years and areas considered, as previously reported (Table 5). 

The 2014 year was the one with the highest level of contamination for all the detected fungal metabolites produced by *Fusarium* spp. of the *Discolor* section, with the exception of CULM, for which 2013 was the most contaminated year. As has already been described for the regulated mycotoxins, DON and ZEA, produced by *Fusarium* spp. of the *Discolor* section and for the DON-associated masked forms, area W was also one of the most contaminated areas for these emerging fungal metabolites, with the exception of NIV, for which no significant differences were recorded between the areas (Table 5). 

A PCA was also carried out to investigate the relationships between the main detected emerging mycotoxins, fungal metabolites, and regulated mycotoxins, as it allowed the maize samples to be clustered for similar mycotoxin contaminations according to the different years and areas. The results of the PCA performed with the regulated and emerging mycotoxins and other fungal metabolites detected in four growing seasons and in five maize areas in Northern Italy were used to produce a loading plot in which the two first principal components again explained 39% and 21% of the total variations of the maize lot samples (Figure 11). The first component differentiated the years (Figure 11A) and the areas (Figure 11B) according to their mycotoxin contents produced by *Fusarium* spp. of the *Discolor* section and was also positively related to the rainfall during the ripening period and negatively related to the mean temperature during the ripening period, thus confirming what had previously been reported for DON and ZEA. Indeed, the contaminations of DON-3-G, 3-ADON, 15-ADON, CULM, AUR, and BUT, which are mainly produced by *F. graminearum* and *F. culmorum* (*Discolor* section), were found to be associated with DON and ZEA (loadings > |0.77|).

The second component instead differentiated the years (Figure 11A) and the areas (Figure 11B) according to the mycotoxin contents produced by *Fusarium* spp. of the *Liseola* section. Indeed, the contaminations of MON, FUS, and BEA, which are mainly produced by *F. subglutinans*, *F. proliferatum*, *F. verticillioides* (*Liseola* section), and *F. avenaceum* (section *Roseum*), were found to be associated with FBs, were positively correlated with the ECB insecticide index (loadings > |0.5|), and were expressed with an arbitrarily assigned number equal to 1 if the maize lot had not been treated with an insecticide, or equal to 0.5 if the lot had instead been treated with a pyrethroid.

## 3. Discussion

This study evaluated, for the first time, mycotoxin profiles obtained from the contamination of regulated, masked, and emerging mycotoxins and fungal metabolites in maize grown in Northern Italy through an extensive survey conducted on maize grain lots from 88 storage centers over the 2011–2021 period, year by year.

When considering the whole area cultivated with maize, the mycotoxin concentrations showed a pronounced year-to-year variation that can be explained by the meteorological conditions during sensitive periods of maize grain development, especially during the ripening period. Several control points, such as the selection and development of resistant cultivars through conventional breeding, Good Agricultural Practices (GAP), chemical and biological control during cultivation, and proper management during storage, may in fact reduce fungal infestation and growth, as well as mycotoxin production in cereal grains. Furthermore, fungal infestation and mycotoxin production depend to a great extent on environmental factors, such as temperature, rainfall, and moisture, at any given stage [17]. 

Almost all the collected samples were contaminated by at least one regulated mycotoxin. A large number of samples (>85%) were co-contaminated with ≥2 mycotoxins. Single species of fungi may actually produce more than one mycotoxin concomitantly, and different fungal species may proliferate and co-exist in the same plant, both circumstances that can lead to the co-occurrence of two or more mycotoxins [18]. Thus, mycotoxin co-contamination is common and is in part governed by the climate; thus, regional trends have emerged throughout the world [7].

The most frequently observed mycotoxin mixtures were combinations of regulated mycotoxins AF B_1_, FBs, as well as DON and ZEA. These results agree with those previously reported by several authors [7,15,17,19,20,21], who noted higher incidences in Southern Europe than in the rest of Europe [7]. It should be highlighted that our data, pertaining to all the detected regulated mycotoxins (AF B_1_, FBs, DON and ZEA), showed much higher incidences and ranges of concentrations than those reported about Europe in several surveys in the literature, as summarized by Lee and Ryu [17], especially for AF B_1_, FBs, and DON. Indeed, the incidences reported by Lee and Ryu [17] for AF B_1_, FBs, and DON were 7%, 49%, and 48%, respectively, while the incidence values recorded over the years in our survey were on average equal to 41%, 98%, and 85%, respectively. Moreover, the ranges and maximum levels at which AF B_1_ (<LOD–145 µg/kg), FBs (< LOD–47,300 µg/kg), and DON (<LOD–36,731 µg/kg) were detected were clearly different from those reported by Lee and Ryu [17] (AF B_1_ = 1.6 µg/kg, FBs = 25–4438 µg/kg and DON = 0.04–2942 µg/kg). Our data confirmed that maize grown in Northern Italy is more prone to contamination than that detected by surveys conducted in other areas.

In addition to the regulated mycotoxins, twelve masked, modified, and emerging mycotoxins and fungal metabolites were also detected in the same maize lots during the in-depth survey conducted over the 2012–2015 period. The co-occurrence of regulated and emerging mycotoxins and fungal metabolites in maize has already been reported throughout Europe for maize for feed and food purposes by several authors [22,23,24,25,26], and our results therefore confirmed the risks related to the presence of emerging mycotoxins and fungal metabolites in food and feeds. 

As far as the relationship between regulated and emerging mycotoxins is concerned, the FB concentrations in the maize lots were correlated with all the emerging mycotoxins and fungal metabolites mainly produced by *Fusarium* spp. section *Liseola* and with ECB damage, thus confirming what has extensively been reported in literature for FBs in North Italian maize surveys [27,28], but which has only been reported for emerging mycotoxins by Blandino et al. in 2015 [29]. Indeed, damage by insects or hail may boost infections because it can create wounds that may readily be colonized by fungi.

On the other hand, the DON and ZEA concentrations in the maize were correlated with all the detected masked forms of DON and all the emerging mycotoxins and fungal metabolites mainly produced by *Fusarium* spp. section *Discolor*. A positive correlation with rainfall and a negative correlation with the average temperatures measured during the ripening period were also observed. Our results are in agreement with those of Gaikpa and Miedaner, who reported that high levels of infections occurred when the weather is cool and wet [30]. Other authors have also reported that *Fusarium* fungi on ears and grains increase under favorable weather conditions, such as higher temperatures and higher rainfall levels in summer, and this results in a longer growing season [31]. Blandino et al. [32] reported that the highest DON and other *Fusarium* spp. *Discolor* section mycotoxin contaminations of maize grain occurred in growing seasons characterized by a high level of precipitation and lower temperatures in the period from maize flowering to harvest.

AF B_1_ was instead detected at the highest levels of contamination in the years and areas characterized by less rainfall and higher temperatures during summer. Indeed, Aspergillus ear rot (AER) has frequently been reported in hot and dry environments, such as Southern USA, Africa, and tropical Asia [33], but has also recently been observed in Southern Europe, especially in Northern Italy, Serbia, Slovenia, Croatia, Romania, and Hungary [34,35]. 

The evaluation of the co-occurrence of the main regulated mycotoxins in maize (AF B_1_, FBs, DON, and ZEA) has also made it possible to define risk areas within the Po Valley. These areas, which are particularly prone to heavy contamination of different mycotoxins, are characterized by different average meteorological conditions, concerning both rainfall and temperatures. Furthermore, these differences could be aggravated by the seasonal meteorological trend. As previously reported, the meteorological conditions in an area are one of the most important key factors for mycotoxin contamination, together with insect damage, to which maize is known to be prone [27,28,29,30,31,32]. Since a correlation has been highlighted between regulated mycotoxins and emerging and masked mycotoxins, which mainly depends on their affinity to the same fungi that produce them, the risk zones/areas described in the present study could be extended to include emerging mycotoxins and fungal metabolites related to the regulated ones.

The definition of risk areas/zones is essential to understand the repercussions of mycotoxin contamination on the food and feed chains, especially in view of a climate change scenario. Indeed, mycotoxins are present at every step of the agri-food chain due to their ability to move through different levels of the food pyramid (carry-over). However, their negative effects are more severe in agricultural crops and dairy farms due to direct losses in harvest, production profitability, animal health, and safety of their products [2,36].

The results of our survey could also be used to identify Good Agricultural Practices (GAP) and structural investments to help rule out mycotoxin contamination. 

In conclusion, this survey not only showed the distribution of mycotoxin contamination in a relevant maize region of the EU but also provided information on the role of the environment on mycotoxins in order to enhance the maize quality of commercial lots, considering the required specifications of the food and feed chains.

## 4. Materials and Methods

### 4.1. Meteorological Data

Climatic parameters, such as the mean rainfall (mm) and mean temperature (°C), were collected during the ripening period (July–September) of maize kernels, the period that appears to be the most closely correlated with the accumulation of mycotoxins. Data were taken from different ARPAE (Agenzia Regionale per la Protezione Ambientale) meteorological stations located in the main Italian maize growing regions. Northern Italy has been geographically divided into five main growing areas: West (W), Center (C), East (E), Adriatic (A), and South Po (SP) to cover the regions with high maize production. Ten meteorological stations were contacted to obtain information: Trino (VC) and Cussanio (CN)—West, Alessandria (AL) and Gonzaga (MN)—South Po River, Bergamo (BG) and Soncino (CR)—Center, Sant’Apollinare (RO) and Ospital Monacale (FE)—Adriatic and Treviso (TV), and Udine (UD)—East. The daily temperatures and precipitation rates were measured at the 10 different stations. Each storage center was connected to the nearest meteorological station, which was always located within a radius of 30 km.

### 4.2. Samples

A total of 3769 representative maize grain samples were collected, after dry processing, over an 11-year period (2011–2021) from 88 storage centers. These storage centers are distributed in five geographical areas (West, Center, South Po, Adriatic, and East), which represent the regions in Northern Italy with the greatest cultivation of maize. A total of 5–10 samples were collected from each storage center each year. A dynamic grain sampling strategy [37] was adopted on the moving product to obtain a representative sample [38]. Accordingly, we took a 15–20 kg pre-sample from all the production components (whole and broken grains and small parts). By applying the standard procedures, sorter samples of 1–1.5 kg each were obtained from these pre-samples for laboratory analyses. These samples were placed in sealed bags and stored in a cool room for 1 day until milling, according to European recommendation [39]. All the samples were subsequently milled to a 1 mm sieve size with a ZM 200 Retsch Ultra-Centrifugal mill equipped with a DR 100 vibratory feeder (Retsch GmbH, Haan, Germany) and stored at 4 °C until chemical analyses.

### 4.3. Regulated Mycotoxins Analyses

The regulated mycotoxin concentration levels in the maize samples collected during the 2011–2021 period were determined by means of an Enzyme-Linked Immunoassorbent Assay (ELISA). Ridascreen^®^ R-Biopharm kit tests (R3401 RIDASCREEN Fumonisin, R1211 RIDASCREEN Aflatoxin B1 30/15, R5906 RIDASCREEN DON, R1401 RIDASCREEN Zearalenon) were performed using Chemwell Automatic Awareness Engineer (inc.). The mycotoxin extraction and tests were performed according to the manufacturers’ instructions. The Limits of Detection (LODs) are declared by the manufacturer: LOD of FBs = 25 µg/kg, LOD of AF B_1_ = 1 µg/kg, LOD of DON = 18.5 µg/kg, and LOD of ZEA = 1.75 µg/kg.

### 4.4. Multi-Mycotoxin LC-MS/MS Analysis

The maize lot samples from the drying and storage centers, collected during the 2012–2015 period, were analyzed by means of a multi-mycotoxin LC-MS/MS analysis. The extraction phase was described in detail by Sulyok et al. in 2006 [40]. The chromatographic and mass spectrometric parameters of the investigated analytes were described by Malachova et al. in 2014 [41]. Briefly, representative 5 g sub-samples of the milled material were extracted using 20 mL of a mixture of acetonitrile/water/acetic acid 79/20/1 (*v*/*v*/*v*). After extraction, the samples were centrifuged, diluted 1 + 1 by a mixture of acetonitrile/water/acetic acid 20/79/1 (*v*/*v*/*v*), and injected.

Detection and quantification were performed with a QTrap 5500 LC–MS/MS System (Applied Biosystems, Foster City, CA, USA), equipped with a TurboIonSpray electrospray ionization (ESI) source and a 1290 Series HPLC System (Agilent, Waldbronn, Germany). Chromatographic separation was performed, at 25 °C, on a Gemini^®^ C18-column, 150 × 4.6 mm i.d., 5 μm particle size, equipped with a C18 security guard cartridge, 4 × 3 mm i.d. (all from Phenomenex, Torrance, CA, USA). Quantification was performed with an external calibration, and the results were corrected for any apparent recoveries determined in the maize. The accuracy of the method was verified by participating in proficiency testing schemes organized by BIPEA (Gennevilliers, France), with 160 out of the 168 results submitted for maize and maize-based feed exhibiting a z-score of between −2 and 2.

### 4.5. Statistical Analysis and Data Elaboration

The Kolmogorov–Smirnov normality test and the Levene test were carried out to verify the normal distribution and the homogeneity of variances, respectively. Since the interaction between the area and year was significant, an analysis of variance (ANOVA) was performed separately for each year with the area as an independent factor for mycotoxin contamination. Multiple comparison tests were performed, by means of the Ryan–Einot–Gabriel–Welsh F (REGW-F) test, on the treatment means. Principal Component Analysis (PCA) was carried out to investigate the relationships between the main detected mycotoxins. All the data obtained were joined together before the PCA was performed with the purpose to reduce the dimensionality of the data set by finding a new set of variables, the two principal components (PC1 and PC2), smaller than the original set of variables, that nonetheless retained most of the sample’s information and the total variance associated. A statistical data analysis was carried out with the SPSS software package, version 24.0.

The data from the monitoring of mycotoxins from the various geo-referenced locations were processed with R software [42] with the *sp, raster, rgdal*, and *leaflet* packages, which is specifically used for geostatistical analysis and geolocalized representation.

The distribution maps were created through the Kriging interpolation process by setting the processing grid with a mesh of 50 × 50 (2500) nodes and a linear function.

## Figures and Tables

**Figure 1 toxins-14-00520-f001:**
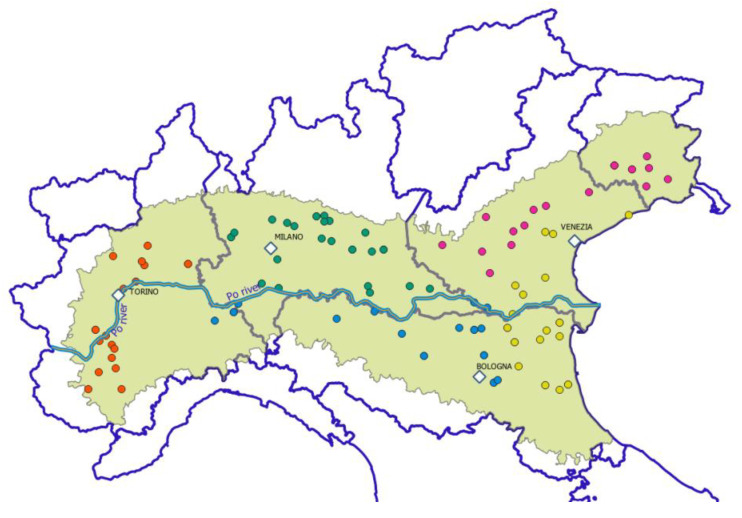
Distribution of the Italian National Maize Monitoring Network in five macro-geographical areas: W, west (Torino) in red; C, center in green; SP, south of the Po River (blue); A, Adriatic (yellow); and E, east (Venezia) in purple.

**Figure 2 toxins-14-00520-f002:**
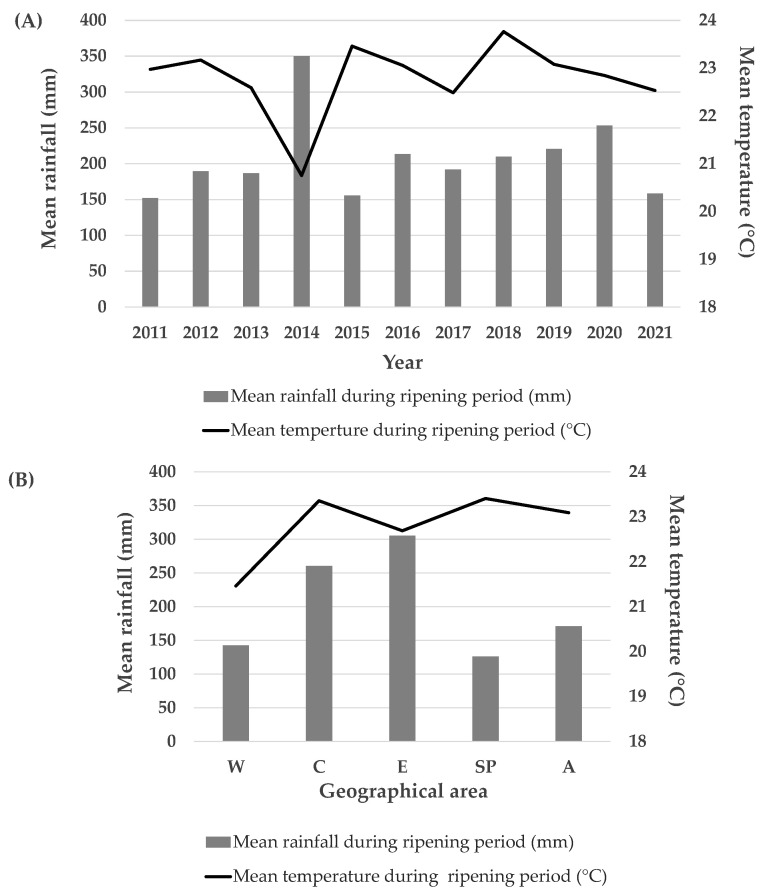
(**A**) Meteorological trend during the 2011–2021 period. (**B**) Meteorological trend in the different investigated geographical maize areas (averaged for 2011–2021 years).

**Figure 3 toxins-14-00520-f003:**
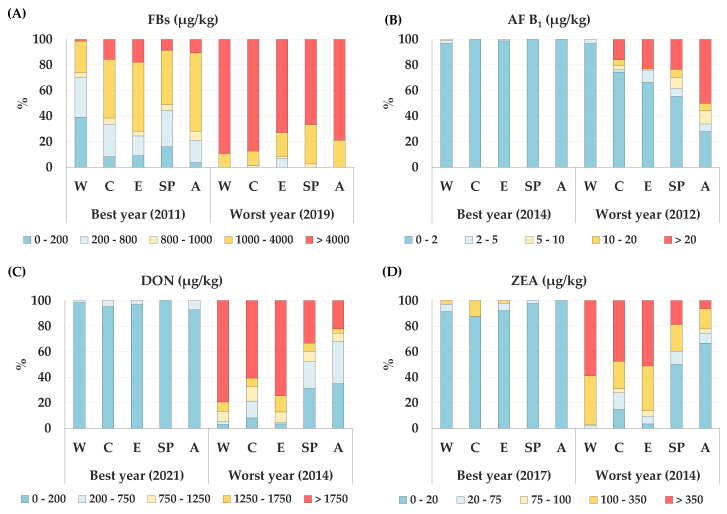
Diffusion of regulated mycotoxins in maize lots from drying and storage centers located in different areas in Northern Italy. For each mycotoxin, the best year with the lowest incidence and contamination was compared with the worst year, with the highest one. Each mycotoxin was expressed as a percentage of incidence over different ranges of increasing contamination, pertaining to Regulations (EC) no. 1126/2007 and no. 165/2010 [3,4]: (**A**) diffusion of FBs; (**B**) diffusion of AF B_1_; (**C**) diffusion of DON; (**D**) diffusion of ZEA.

**Figure 4 toxins-14-00520-f004:**
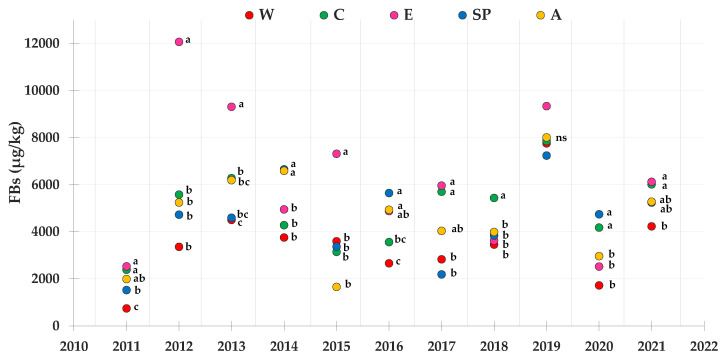
Contamination levels of FBs referring to the average toxin levels found in maize lots from drying and storage centers located in different areas in Northern Italy during the 2011–2021 period. Means followed by different letters are significantly different (*p*-value < 0.05), according to the REGW-F test, conducted separately for each year (ns = not significant, that is, the different areas were not statistically different from each other as far as the FB contamination level is concerned).

**Figure 5 toxins-14-00520-f005:**
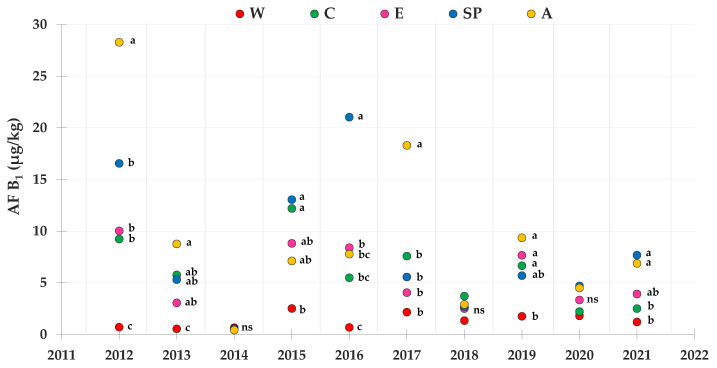
Contamination levels of AF B_1_ referring to the average toxin levels found in maize lots from drying and storage centers located in different areas in Northern Italy during the 2012–2021 period. Means followed by different letters are significantly different (*p*-value < 0.05), according to the REGW-F test, conducted separately for each year (ns = not significant, that is, the different areas were not statistically different from each other as far as the AF B_1_ contamination level is concerned).

**Figure 6 toxins-14-00520-f006:**
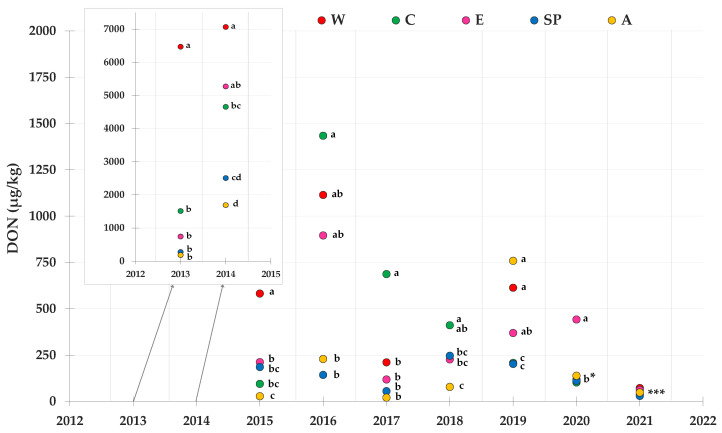
Contamination levels of DON referring to the average toxin levels found in maize lots from drying and storage centers located in different areas in Northern Italy during the 2013–2021 period. Means followed by different letters are significantly different (*p*-value < 0.05), according to the REGW-F test, conducted separately for each year (ns = not significant, that is, the different areas were not statistically different from each other as far as the DON contamination level is concerned; b * = the areas A, W, SP and C were not statistically different from each other and they were assigned the letter b; *** = the areas were significantly different (*p*-value < 0.05) and they were assigned different letters as follows: area W→a; areas E and A→ab; areas C and SP→b).

**Figure 7 toxins-14-00520-f007:**
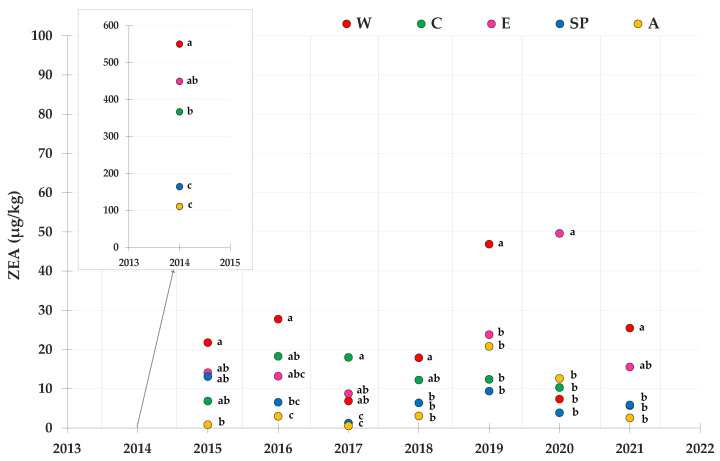
Contamination levels of ZEA referring to the average toxin levels found in maize lots from drying and storage centers located in different areas in Northern Italy during the 2014–2021 period. Means followed by different letters are significantly different (*p*-value < 0.05), according to the REGW-F test, conducted separately for each year (ns = not significant, that is, the different areas were not statistically different from each other as far as the ZEA contamination level is concerned).

**Figure 8 toxins-14-00520-f008:**
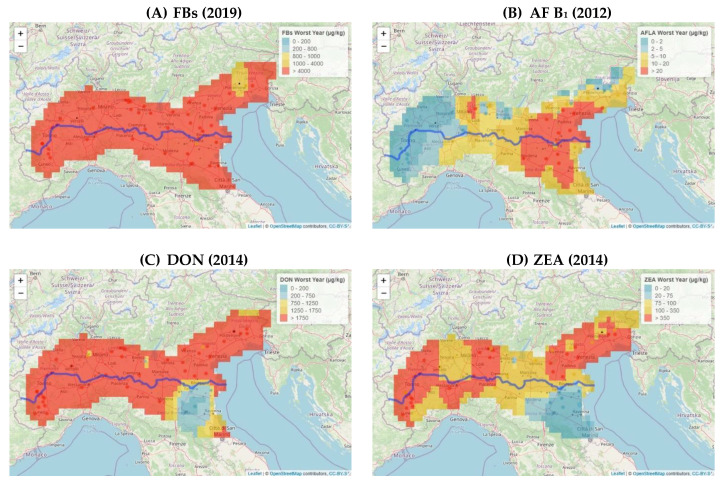
Diffusion of regulated mycotoxins in maize lots from drying and storage centers located in the Po Valley. The worst year, that is, the one with the highest incidence and contamination, was expressed for each mycotoxin as a percentage of incidence over different ranges of increasing contamination, pertaining to Regulations (EC) no. 1126/2007 and no. 165/2010 [3,4]: (**A**) diffusion of FBs; (**B**) diffusion of AF B_1_; (**C**) diffusion of DON; (**D**) diffusion ZEA.

**Figure 9 toxins-14-00520-f009:**
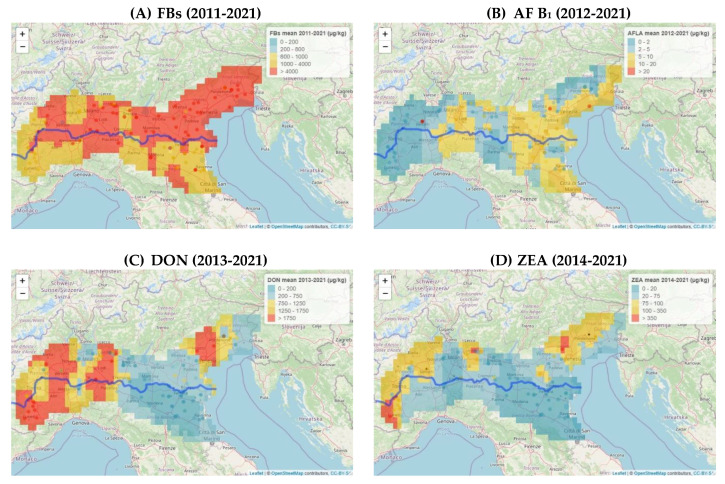
Diffusion of regulated mycotoxins in maize lots from drying and storage centers located in the Po Valley. The mean concentration, detected during the investigated years, was expressed for each mycotoxin as a percentage of incidence over different ranges of increasing contamination, pertaining to Regulations (EC) no. 1126/2007 and no. 165/2010 [3,4]: (**A**) diffusion of FBs; (**B**) diffusion of AF B_1_; (**C**) diffusion of DON; (**D**) diffusion of ZEA.

**Figure 10 toxins-14-00520-f010:**
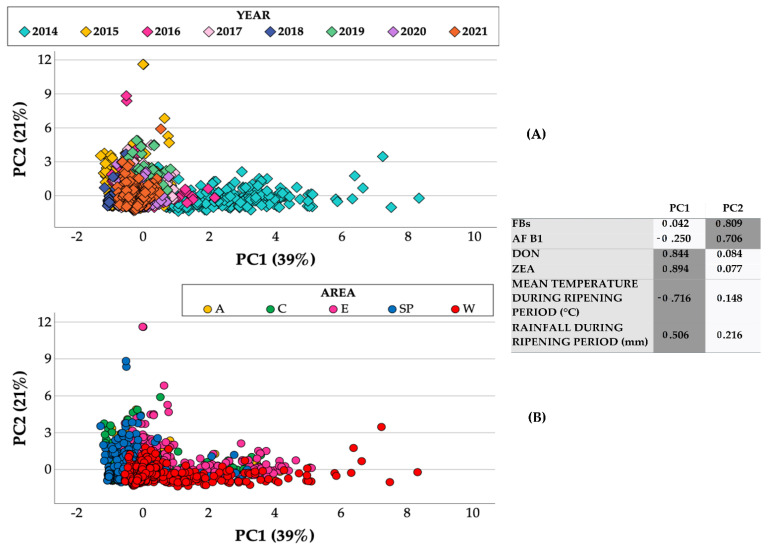
Score plots of the considered maize lots (N = 3769) from drying and storage centers located in different areas in Northern Italy during the 2014–2021 period, marked according to the year (**A**) and area (**B**). The loadings (highlighted values > |0.5|) of each parameter (regulated mycotoxins and meteorological parameters) are reported with the two first principal components.

**Figure 11 toxins-14-00520-f011:**
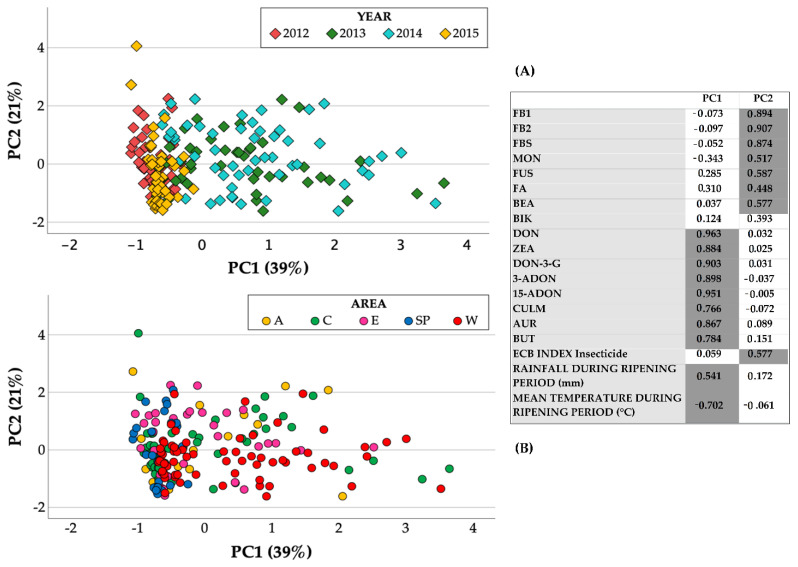
Score plots of the maize lots (N = 200) from drying and storage centers located in different areas in Northern Italy during the 2012–2015 period, marked according to the year (**A**) and area (**B**). The loadings (highlighted values > |0.5|) of each parameter (masked and emerging mycotoxins, other fungal metabolites, meteorological and insect parameters) are reported with the two first principal components.

**Table 1 toxins-14-00520-t001:** The occurrence of FBs and AF B_1_ in maize lots from drying and storage centers.

Factor	Source of Variation	Samples (No.)	FBs			AF B_1_		
			Average (µg/kg)	Min–Max (µg/kg)	Pos *** (%)	Average (µg/kg)	Min–Max (µg/kg)	Pos *** (%)
**Year (Y)**	2011	456	1886 ^g^	<LOD ****–18,990	97	−	−	−
	2012	362	6685 ^b^	<LOD ****–47,300	97	12.5 ^a^	<LOD *****–115	45
	2013	388	6447 ^b^	<LOD ****—27,000	99	4.4 ^cde^	<LOD *****—119	22
	2014	355	5035 ^cd^	<LOD ****—28,920	100	0.6 ^f^	<LOD *****–7	7
	2015	384	3815 ^ef^	<LOD ****–38,260	98	8.7 ^b^	<LOD *****–145	48
	2016	314	4157 ^def^	<LOD ****–23,220	99	7.6 ^b^	<LOD *****–145	61
	2017	259	4212 ^de^	80–21,270	100	6.4 ^bc^	<LOD *****–91	50
	2018	319	4209 ^de^	<LOD ****–16,280	99	2.7 ^ef^	<LOD *****–78	36
	2019	307	8073 ^a^	320—23,600	100	6.2 ^bcd^	<LOD *****–62	59
	2020	332	3224 ^f^	<LOD ****–19,590	98	3.1 ^def^	<LOD *****–75	45
	2021	293	5330 ^c^	<LOD ****–29,852	97	4.0 ^cde^	<LOD *****–51	34
	*p*-value	−	**<0.001**	−	−	**<0.001**	−	−
**Area (A)**	W	793^FBs/^716^AF B_1_^	3507 ^d^	<LOD ****–28,920	99	1.3 ^c^	<LOD *****–44	18
	C	990^FBs/^868^AF B_1_^	4820 ^b^	<LOD ****–29,852	99	5.7 ^b^	<LOD *****–119	41
	E	793^FBs/^716^AF B_1_^	6577 ^a^	<LOD ****–47,300	98	5.3 ^b^	<LOD *****–145	39
	SP	566^FBs/^472^AF B_1_^	4206 ^c^	<LOD ****–24,360	97	8.3 ^a^	<LOD *****–145	52
	A	751^FBs/^673^AF B_1_^	4500 ^bc^	<LOD ****–22,830	99	9.2 ^a^	<LOD *****–117	56
	*p*-value	-	**<0.001**	-	-	**<0.001**	-	-
**Y × A**	*p*-value	-	**<0.001**	-	-	**<0.001**	-	-

Means followed by different letters, reported in superscript, are significantly different (the level of significance of the *p*-value is reported in the table), according to the REGW-F test. *** Percentage of samples above the Limit of Detection (LOD). **** LOD of FBs = 25 µg/kg. ***** LOD of AF B_1_ = 1 µg/kg.

**Table 2 toxins-14-00520-t002:** The occurrence of DON and ZEA in maize lots from drying and storage centers.

Factor	Source of Variation	Samples (No.)	DON			ZEA		
			Average (µg/kg)	Min–Max (µg/kg)	Pos *** (%)	Average (µg/kg)	Min–Max (µg/kg)	Pos *** (%)
**Year (Y)**	2013	388	2049 ^b^	<LOD ****–36,731	94	−	−	−
	2014	355	4646 ^a^	50–36,583	100	364 ^a^	<LOD *****–1395	91
	2015	384	221 ^d^	<LOD ****–2740	77	11 ^b^	<LOD *****–315	31
	2016	314	917 ^c^	<LOD ****–17,236	94	15 ^b^	<LOD *****–252	54
	2017	259	264 ^d^	<LOD ****–7623	75	8 ^b^	<LOD *****–175	25
	2018	319	336 ^d^	<LOD ****–3865	82	10 ^b^	<LOD *****–158	44
	2019	307	426 ^cd^	<LOD ****–9064	93	23 ^b^	<LOD *****–324	66
	2020	332	166 ^d^	<LOD ****–2903	85	15 ^b^	<LOD *****–258	42
	2021	293	50 ^d^	<LOD ****–375	65	11 ^b^	<LOD *****–185	36
	*p*-value	−	**< 0.001**	−	−	**< 0.001**	−	−
**Area (A)**	W	655^DON/^568^ZEA^	2232 ^a^	<LOD ****–36,731	97	112 ^a^	<LOD *****–1395	70
	C	774^DON/^686^ZEA^	895 ^b^	<LOD ****–24,407	84	43 ^b^	<LOD *****–1142	45
	E	581^DON/^481^ZEA^	1148 ^b^	<LOD ****–26,064	88	95 ^a^	<LOD *****–1181	54
	SP	425^DON/^378^ZEA^	424 ^c^	<LOD ****–13,325	79	27 ^bc^	<LOD *****–840	35
	A	516^DON/^450^ZEA^	372 ^c^	<LOD ****–13,925	72	21 ^c^	<LOD *****–1280	34
	*p*-value	−	**<0.001**	−	−	**<0.001**	−	−
**Y × A**	*p*-value	−	**<0.001**	−	−	**<0.001**	−	−

Means followed by different letters, reported in superscript, are significantly different (the level of significance of the *p*-value is reported in the table), according to the REGW-F test. *** Percentage of samples above the Limit of Detection (LOD). **** LOD of DON = 18.5 µg/kg. ***** LOD of ZEA = 1.75 µg/kg.

**Table 3 toxins-14-00520-t003:** The occurrence of the masked mycotoxins DON-3-G, 3-ADON, and 15-ADON in maize lots from drying and storage centers.

Factor	Source of Variation	DON-3-G		3-ADON		15-ADON	
		Average (µg/kg)	Pos *** (%)	Average (µg/kg)	Pos *** (%)	Average (µg/kg)	Pos *** (%)
**Year (Y)**	2012	133 ^c^	89	5 ^c^	100	26 ^c^	100
	2013	595 ^b^	100	87 ^a^	100	290 ^b^	100
	2014	1247 ^a^	100	64 ^b^	91	444 ^a^	94
	2015	54 ^c^	60	2 ^c^	6	25 ^c^	23
	*p*-value	**<0.001**	-	**<0.001**	-	**<0.001**	-
**Area (A)**	W	791 ^a^	100	68 ^a^	80	315 ^a^	89
	C	469 ^b^	85	39 ^b^	75	206 ^b^	73
	E	476 ^b^	79	21 ^bc^	67	155 ^bc^	64
	SP	33 ^c^	78	5 ^c^	52	22 ^c^	70
	A	466 ^b^	78	22 ^bc^	78	124 ^bc^	87
	*p*-value	**<0.001**	-	**<0.001**	-	**<0.001**	-
**Y × A**	*p*-value	**<0.001**	-	**<0.001**	-	**<0.001**	-

Means followed by different letters, reported in superscript, are significantly different (the level of significance of the *p*-value is reported in the table), according to the REGW-F test. *** Percentage of samples above the LOD (DON-3-G = 0.8 µg/kg; 3-ADON = 1.2 µg/kg; 15-ADON = 12 µg/kg).

**Table 4 toxins-14-00520-t004:** The occurrence of the emerging mycotoxins and fungal metabolites mainly produced by *Fusarium* spp. section *Liseola* (BEA, BIK, FA, FUS, and MON) in maize lots during the 2012–2015 period.

Factor	Source of Variation	BEA		BIK		FA		FUS		MON	
		Average (µg/kg)	Pos *** (%)	Average (µg/kg)	Pos *** (%)	Average (µg/kg)	Pos *** (%)	Average (µg/kg)	Pos *** (%)	Average (µg/kg)	Pos *** (%)
**Year (Y)**	2012	187 ^ab^	100	294 ^b^	100	356 ^bc^	100	959 ^bc^	100	852 ^a^	100
	2013	195 ^a^	100	853 ^a^	100	1236 ^a^	100	1551 ^a^	100	344 ^c^	100
	2014	135 ^bc^	100	175 ^c^	100	483 ^b^	98	1321 ^ab^	98	505 ^b^	100
	2015	101 ^c^	100	102 ^c^	100	201 ^c^	87	697 ^c^	79	574 ^b^	100
	*p*-value	**<0.001**	-	**<0.001**	-	**<0.001**	-	**0.003**	-	**<0.001**	-
**Area (A)**	W	189 ^a^	100	306 ^b^	100	492 ^bc^	97	1187 ^a^	100	554 ^a^	100
	C	131 ^a^	100	335 ^ab^	100	604 ^ab^	96	1423 ^a^	98	622 ^a^	100
	E	126 ^a^	100	394 ^ab^	100	588 ^abc^	97	848 ^a^	88	468 ^a^	100
	SP	144 ^a^	100	277 ^b^	100	344 ^c^	89	835 ^a^	85	686 ^a^	100
	A	144 ^a^	100	470 ^a^	100	825 ^a^	100	1011 ^a^	87	482 ^a^	100
	*p*-value	0.196	-	**<0.001**	-	**<0.001**	-	0.075	-	0.190	-
**Y × A**	*p*-value	0.136	-	**<0.001**	-	**<0.001**	-	**0.040**	-	**0.011**	-

Means followed by different letters, reported in superscript, are significantly different (the level of significance of the *p*-value is reported in the table), according to the REGW-F test. *** Percentage of samples above the LOD (BEA = 0.008 µg/kg; BIK = 8 µg/kg; FA = 16 µg/kg; FUS = 40 µg/kg; MON = 1.6 µg/kg).

**Table 5 toxins-14-00520-t005:** The occurrence of the emerging mycotoxins and fungal metabolites mainly produced by *Fusarium* spp. section *Discolor* (AUR, BUT, CULM, and NIV) in maize lots from drying and storage centers.

Factor	Source of Variation	AUR		BUT		CULM		NIV	
		Average (µg/kg)	Pos *** (%)	Average (µg/kg)	Pos *** (%)	Average (µg/kg)	Pos *** (%)	Average (µg/kg)	Pos *** (%)
**Year (Y)**	2012	161 ^c^	77	41 ^b^	66	109 ^c^	53	2 ^b^	100
	2013	3929 ^b^	100	567 ^a^	94	2438 ^a^	100	21 ^b^	100
	2014	9642 ^a^	100	592 ^a^	100	970 ^b^	100	49 ^a^	89
	2015	180 ^c^	75	89 ^b^	75	58 ^c^	60	5 ^b^	11
	*p*-value	**<0.001**	-	**<0.001**	-	**<0.001**	-	**<0.001**	-
**Area (A)**	W	4620 ^a^	98	390 ^a^	100	1595 ^a^	100	15 ^a^	75
	C	4512 ^a^	88	337 ^a^	85	1037 ^b^	71	14 ^a^	75
	E	3213 ^ab^	85	355 ^a^	79	298 ^c^	79	19 ^a^	70
	SP	481 ^c^	74	37 ^b^	56	42 ^c^	52	35 ^a^	67
	A	2561 ^ab^	78	393 ^a^	78	241 ^c^	65	28 ^a^	78
	*p*-value	**<0.001**	-	**<0.001**	-	**<0.001**	-	0.383	-
**Y × A**	*p*-value	**0.002**	-	**0.005**	-	**<0.001**	-	0.282	-

Means followed by different letters, reported in superscript, are significantly different (the level of significance of the *p*-value is reported in the table), according to the REGW-F test. *** Percentage of samples above the LOD (AUR = 2.4 µg/kg; BUT = 5.6 µg/kg; CULM = 8 µg/kg; NIV = 1.2 µg/kg).

## Data Availability

The data that support the findings of this study are available from the corresponding author, [S.L.], upon reasonable request.

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
