# Peer review of "Multi-Mycotoxin Long-Term Monitoring Survey on North-Italian Maize over an 11-Year Period (2011–2021): The Co-Occurrence of Regulated, Masked and Emerging Mycotoxins and Fungal Metabolites"

_toxins, 2022, doi:10.3390/toxins14080520_

Round 1
Reviewer 1 Report
The authors tried to systemize the data on maize contaminations by mycotoxins in several Italian areas during 2011-2021 years. The results confirmed well-known trends in similar pollution worldwide. This manuscript can be interesting mostly agricultural specialists owing to the duration of the statistical observations made.
I would like to give some comments to the authors:
Abstract:
- Now, Abstract is twice lager as it should be: “A single paragraph of about 200 words maximum.” So, it should be generally modified.
- Please indicate the abbreviation “DOP" if it remains in the text of the abstract in the new edition.
Results:
Lines 90-94: The text “This network, which is coordinated by the CREA Research Centre for Cereal and Industrial Crops, is composed of 88 maize storage centers distributed throughout five macro-geographical areas, which, in Figure 1, are indicated with different colors: West (W) in red, Center (C) in green, South of the Po River (SP) in blue, Adriatic (A) in yellow, and East in purple.” should be changed to “This network, which is coordinated by the CREA Research Centre for Cereal and Industrial Crops, is composed of 88 maize storage centers distributed throughout five macro-geographical areas (Figure 1). The Po Valley is the main maize growing area in Italy.”
The further text “The main maize growing area in Italy is the Po Valley (Figure 1, green area), which is in the north of the peninsula. It extends from West (Turin) to East (Venice) and above and below the course of the Po River.” should be deleted, since that is clear from the Figure 1.
Figure 1: the title should be changed to following version: “Figure 1. Distribution of the Italian National Maize Monitoring Network in five macro-geographical areas: West (Turin) in red, Center (Po Valley) in green, South of the Po River in blue, Adriatic in yellow, and East (Venice) in purple.”
Figure 2: please, add the titles of axes and the quality of the picture should be improved (not all numbers are well visualized).
Figure 3: please, add the titles of axes and change the title to “Meteorological trend in the different investigated geographical maize areas (W = West; C 127 = Center; E = East; SP = South Po; A = Adriatic) averaged for 2011-2021 years”.
Section 2.3: please delete from the text all the data (concentrations) that are presented in Table 1.
Line 152: The abbreviation for deoxynivalenol (DON) and zearalenone (ZEA) were introduced already earlier (line 50).
Same situation is in titles of Fig.4, Fig.9 and Fig.10: “(C) diffusion of deoxynivalenol (DON); (D) diffusion of zearalenone (ZEA).” It is not necessary to introduce same abbreviations many times to the manuscript.
Figures 5- 7: Please add the explanations of all letters to the Titles of the figures! If area W → a; areas E and A → ab; areas C and SP → b (Fig.5), so please, explain the meaning of letters “c, d, bc, and cd”. What is it? Sorry, the pictures are not clear.
Figures 11 and 12: Please, explain: what is it PC1 (39%) and PC2 (21%)(axes)? It is absolutely unclear. There are no explanations of the calculations in part of Materials &Methods.
Lines 439-440: There is a doubling of the information provided by the names and numbers of the mentioned links, so the text “Marin et al. (2013), Lee and Ryu (2017), Gruber-Dorninger et al. (2019), Escola et al. ((2020), Hodai et al. (2021) and Weaver et al. (2021)” should be deleted.
General recommendations: the text is hard to read, as there are a lot of repetitions and numbers in it. So, I recommend removing all the doubling of thoughts from the text and conducting a deeper systematization of the data.
Reviewer 2 Report
The manuscript entitled “Multi-mycotoxin long-term monitoring survey on North-Italian maize over an 11-year period (2011-2021): the co-occurrence of regulated, masked and emerging mycotoxins and fungal metabolites” is significant in this field of interest. The manuscript is well-structured with enough data. However, this manuscript has a few questions needed to be addressed. Thus I recommend this manuscript for minor revision.
Ø All tables’ titles are too long. Kindly shorten the title, which fits into a single line.
For eg:
Ø Table 1. The occurrence of fumonisins B and aflatoxin B1 in maize lots from drying and storage centers.
Ø Provide remaining information as table footnote/legend--- fumonisins B (FBs = sum of fumonisin B1, FB1 and fumonisin B2, FB2), aflatoxin B1 (AF B1), (W = West; C = Center; E = East; SP = South Po; A = Adriatic)
Ø Table 4. The occurrence of the emerging mycotoxins and fungal metabolites in Northern Italy during 2012-2015
Ø The following should be in the table footnote/legend (beauvericin, BEA; bikaverin, BIK; fusaric acid, FA; fusaproliferin, FUS and moniliformin, MON; W = West; C = Center; E = East; SP = South Po; A = Adriatic)
Ø In table 1 please check the average values (alphabets) and should be in superscript.
Ø Please include what this is an indication of these alphabets on average values.
Ø What about the LOD value for 2017 and 2019?
Ø In table 1 if all LOD of fumonisins (FBs) is 25 µg/kg please mention it in the initial occurrence only, not in all places. The same for LOD of aflatoxin B1.
Ø Table 4. The occurrence of the emerging mycotoxins and fungal metabolites in Northern Italy during 2012-2015
Ø The following should be in table legend (beauvericin, BEA; bikaverin, BIK; fusaric acid, FA; fusaproliferin, FUS and moniliformin, MON; W = West; C = Center; E = East; SP = South Po; A = Adriatic)
Ø In table 4 kindly mention the Limit of Detection (LOD) of all toxins.
Ø As MDPI (toxins) journals are open access, most readers use softcopy of articles to read; hence the color figures will increase the clarity of the figure. Thus if possible, kindly replace all figures (with color one).
Ø In figure 5, kindly provide mean + SD values for better understanding. For example, in 2019, even though there is a difference in the mean value E = East with other places, it indicates insignificant. Here without SD values, it is not clear and follows this for all figures.
Ø Why do the authors use five decimal values in Figures 11 and 12 scales, I hope if 0, 1, 2, 3, etc.. will make the figures better readability.
Kindly use the present article data to add more information and correlation about how mycotoxin production is affected by environmental factors, such as temperature, rainfall, and moisture.
Reviewer 3 Report
This study investigated the occurrence of multi-mycotoxin on maize collected from the north of Italy between 2011 and 2021. The work can clarify the contamination level of major mycotoxins as well as the correlation between mycotoxins production and the local meteorological conditions, which provide useful information for mycotoxin control in field management. I recommend accepting this paper after some modifications.
Abstract: the description of this part is so fussy. Please simplify the section. In addition, what are the masked, emerging mycotoxins and fungal metabolites? They should be mentioned in the Abstract section.
Line 22: fungal metabolites
Figure 2 and Figure 3 could be integrated into one picture
Table 1 and Table 2: the maize sample amounts collected from different years or locations should be present in the table 1. The different letters should be superscript. Besides, how did the authors obtain the LODs of FBs and AFB1? ELISA method or LC-MS/MS method?
Line 327-328: In my opinion, 3-ADON and 15-ADON belong to the regulated mycotoxins rather than masked toxins. Please clarify this question.
Line 513-514: Please add the reference literature.
Line 540: Which mycotoxins were tested using the ELISA method? The method sensitivity and precision should be shown in this part.
Line 545: As mentioned in manuscript, the maize samples collected during the 2012-2015 year were analyzed by LC-MS/MS method. Was the ELISA method employed to determine the samples from the years 2011 and 2016-2021? Were all the ELISA kits of these mycotoxins commercially available? In addition, the analytical method based on LC-MS/MS should be described including the sample pretreatment and instrumental conditions. That’s very important for the quantification results.
Round 2
Reviewer 1 Report
I highly appreciate the work with the manuscript, which was carried out by its authors, I agree with most of the changes made to the text, but there were minor comments:
Please, look at the captions to Figures 4,5,6 and 7: I'm sorry, but it remains unclear what the authors indicated in the figures with the letters “c” and “d". In the case of Figures 4, 5 and 7, there are no explanations of all letters at all, there is a partial explanation only in the caption to Figure 6. Firstly, all explanations should be given under each Figure, which should be understandable without comparing the captions with neighboring Figures.
Secondly, I would like to ask the authors once again to look very carefully at their Figures, especially at the "inserts" in Figures 6 and 7, where these letters (“c” and “d") are present. Here we can see the caption with the following text: "b* = the areas A, W, SP and C were not statistically different from each other and they were assigned the letter b; *** = the areas were significantly different (p-value < 0.05) and they were assigned different letters as follows: area W → a; areas E and A → ab; areas C and SP → b).” I do not see the letters “c” and “d" here, but there are also combinations of letters “c”, “d”, “bc”, “abc”, “cd". Are these also some kind of “areas"? Which ones, I don't see the explanations. Again, there are only explanations for the letters a, ab, b. Sorry, without all the necessary disclosures of the designations, the Figures are not acceptable.
The authors gave the following answer to my question in the previous round of examination:
Figures 11 and 12: Please, explain: what is it PC1 (39%) and PC2 (21%) (axes)? It is absolutely unclear. There are no explanations of the calculations in part of Materials &Methods.
In the Materials & Methods we reported: “Principal Component Analysis (PCA) was carried out to investigate the relationships between the main detected mycotoxins. All the data obtained were joined together be-fore the PCA was performed.”. PC1 and PC2 are respectively the first and second principal components, obtained after the application of the PCA statistical analysis, which explain respectively 39% and 21% of the total variance associated with the variables included in the test and this test allows to reduce the number of variables to only 2, the 2 principal components.”
Unfortunately, there are no such explanations in Materials & Methods. I didn't find them there, although they could have been added to the Materials & Methods section, part 4.5, Line 550. However, there is following text in the manuscript (Lines 271-274):"The results of the PCA performed with the regulated mycotoxins detected over eleven growing seasons and in five maize areas in North-Italy were used to produce a loading plot in which the two first principal components explained 39% and 21% of the total variations of the maize lot samples (Figure 10).", but it is not clear from this text what PC1 and PC2 are. It is necessary to add this explanation here as well.
